# Prevalence and determinants of wasting of under-5 children in Bangladesh: Quantile regression approach

**Md. Moyazzem Hossain**[1,2]*, **Faruq Abdulla**[3]*, **Azizur Rahman**[4]

**1** Department of Statistics, Jahangirnagar University, Savar, Dhaka, Bangladesh, **2** School of Mathematics, Statistics & Physics, Newcastle University, Newcastle upon Tyne, United Kingdom, **3** Department of Applied Health and Nutrition, RTM Al Kabir Technical University (RTM-AKTU), Sylhet, Bangladesh, **4** School of Computing, Mathematics and Engineering, Charles Sturt University, Wagga Wagga, New South Wales, Australia

* hossainmm@juniv.edu (MMH); faruqiustat09mnil@gmail.com (FA)

**Data Availability Statement:** The access link of data set is http://dhsprogram.com/data/available-datasets.cfm.

## Abstract

### Background

Wasting is perhaps one of the signs of malnutrition that has been linked to the deaths of children suffering from malnutrition. As a result, understanding its correlations and drivers is critical. Using quantile regression analysis, this research aims to contribute to the discussion on under-5 malnutrition by analyzing the predictors of wasting in Bangladesh.

### Methods and materials

The dataset was extracted from the 2017–18 Bangladesh demographic and health survey (BDHS) data. The weight-for-height (WHZ) z-score based anthropometric indicator was used in the study as the target variable. The weighted sample constitutes 8,334 children of under-5 years. However, after cleaning the missing values, the analysis is based on 8,321 children. Sequential quantile regression was used for finding the contributing factors.

### Results

The findings of this study depict that the prevalence of wasting in children is about 8 percent and only approximately one percent of children are severely wasted in Bangladesh. Age, mother's BMI, and parental educational qualification, are all major factors of the WHZ score of a child. The coefficient of the female child increased from 0.1 to 0.2 quantiles before dropping to 0.75 quantile. For a child aged up to three years, the coefficients have a declining tendency up to the 0.5 quantile, then an increasing trend. Children who come from the richest households had 16.3%, 3.6%, and 15.7% higher WHZ scores respectively than children come from the poorest households suggesting that the risk of severe wasting in children under the age of five was lower in children from the wealthiest families than in children from the poorest families. The long-term malnutrition indicator (wasting) will be influenced by the presence of various childhood infections and vaccinations. Furthermore, a family's economic position is a key determinant in influencing a child's WHZ score.

**Funding:** The author(s) received no specific funding for this work.

**Competing interests:** The authors have declared that no competing interests exist.

## Conclusions

It is concluded that socioeconomic characteristics are correlated with the wasting status of a child. Maternal characteristics also played an important role to reduce the burden of malnutrition. Thus, maternal nutritional awareness might reduce the risk of malnutrition in children. Moreover, the findings disclose that to enrich the nutritional status of children along with achieving Sustainable Development Goal (SDG)-3 by 2030, a collaborative approach should necessarily be taken by the government of Bangladesh, and non-governmental organizations (NGOs) at the community level in Bangladesh.

## Introduction

Childhood undernutrition has long-term consequences such as reduced attainment of schooling, lessened economic potential, and chronic illness in adulthood. It is more common in underdeveloped nations and is linked to child fatalities. Therefore, as we move forward with the Sustainable Development Goals (SDGs), we should promote and maintain nutritional wellbeing. Malnutrition is acting as a leading cause of death among under-5 children, as well as one of the most widespread causes of deterioration in children's health and well-being, resulting in impaired learnability, incompetence, and inefficiency to acquire skills [1]. Childhood malnutrition is a major contributor to the global burden of illness and a primary cause of mortality and morbidity observed among under-5 children in poor and middle-income countries like Bangladesh [2–4]. Moreover, nutritional deficiencies raise the likelihood of death from frequent infections, the prevalence of infections, and may cause infection recovery to be prolonged [4–6].

Among the indicators of malnutrition, wasting is one of the signs of malnutrition that has been linked to the deaths of children suffering from malnutrition [7, 8]. Wasting decreased overall across low-and middle-income countries (LMICs) between 2000 and 2017, from 8.4% to 6.4%, however, it is still above the World Health Organization's Global Nutrition Target of less than 5% [9]. A previous study mentioned that the prevalence of wasting is 15.49% in district Rahimyar Khan, Pakistan [10] and the probability of child malnutrition was lower among the children of mothers who had high mothers' nutritional and health awareness [11]. In 2020, more than 45 million of under-5 children were influenced by wasting, among them 13.6 million children have been severely wasted. However, these figures are impacted by COVID-19 and because of degradation in household wealth and interruptions in the availability and cost of nutritious food as well as vital nutrition services, it is anticipated that 1.15 times more children will be impacted by wasting in 2020 than previously estimated [12].

The nutritional status of a country's children is a barometer of its socio-economic development. Poor nutritional status, on the other hand, is one of Bangladesh's most profound health as well as welfare issues. Bangladesh has made strenuous efforts to minimize child malnutrition and has had some results. According to BDHS2017-18 reports, the level of wasting decreased by about half of the previous years [13]. Several studies have found a link between childhood malnutrition and a variety of factors such as an individual's socioeconomic status, demographic features, environmental factors, household factors, parental attributes, child-feeding habits, child morbidity, vitamin intake and vaccination coverage, geographic location, and residency [14–29].

Previous studies, on the other hand, have largely looked at the predictors using multiple linear regression or logistic regression models. However, at different points, the mean effect may overestimate or underestimate the contribution of the covariates. Another notable drawback of logistic regression models is that they assess observations that are below or above a cut-off point equally, ignoring the magnitude of deviations from that threshold level. As a result, statistical information that could be useful for intervention along with health promotion initiatives could be lost. However, the quantile regression (QR) model has the benefit that it is robust in the preference of outliers [30]. Previous studies highlighted that mean and variance are affected by outliers and proposed mean and variance in the presence of outliers [31, 32]. In addition, the QR model yielded more unbiased estimates for skewed data than the linear regression model [33]. Previous studies applied the QR model for exploring the contributing factor of the age of the mother at first birth and age at first marriage because of its non-normality nature [34–36]. Moreover, several researchers applied the QR model to examine the core socio-demographic factors of child nutritional status [37–41]. A prior study used a simultaneous quantile regression model to identify significant risk factors for severe stunting in children under the age of five [42]. Researchers also explored the risk factors linked with malnutrition among under-5 children using the multilevel, spatial-temporal model, and geostatistical analysis [43–46].

The authors are well known about the three different dimensions used indicators of nutritional status of under-5 children. Stunting measures chronic nutritional deficiency, wasting is a measure of acute nutritional deficiency, and underweight is a composite measure of both acute and chronic statuses. According to a prior study, there are 45.4 million wasted children under the age of five [47, 48]. Despite the fact that the global prevalence of wasting has gradually decreased, however, only more than a quarter of 194 countries are on track to meet the World Health Assembly's (WHA) 2025 target of keeping the prevalence of wasting under 5.0 percent [48, 49]. It has the greatest short-term case fatality rate of any form of malnutrition [50, 51]. There's also evidence that wasting can be a 'harbinger of stunting,' with episodes of wasting impairing linear growth [52]. Hence, the skewness coefficient, and the Boxplot of the WHZ-scores investigated in this study, shown in Fig 1, demonstrate the existence of outliers and their distribution does not fully match the normal distribution. Moreover, the authors do not find any study on wasted children based on BDHS-2017/18 data considering QR regression. These reasons are working behind for using the QR model to explore the contributing predictors of wasting among children aged less than 5 years in Bangladesh considering the most recent BDHS-2017-18 data.

## Methods and materials

### Data and variables

In this study, the secondary data is obtained from a nationally representative survey called the 2017–18 Bangladesh Demographic and Health Survey (BDHS-2017/18). The BDHS-2017-18 is the complete survey that covers the enumeration areas (EAs) of the entire country. This survey used stratified sampling and selection is made in two stages. Firstly, 675 EAs were chosen with probability proportional to the size of the EA. In the second phase of selection, 30 households per cluster were carefully chosen with a systematic procedure from the list of households. However, due to natural disasters, data were not collected from 3 EAs. These three clusters were in Rajshahi (one rural cluster), Rangpur (one rural cluster), and Dhaka (one urban cluster). The full data set is accessible via the following link http://dhsprogram.com/data/available-datasets.cfm. Before starting the analysis the authors use a weighted sample to make sure the country representative sample. The details of the sampling procedure and methods of the

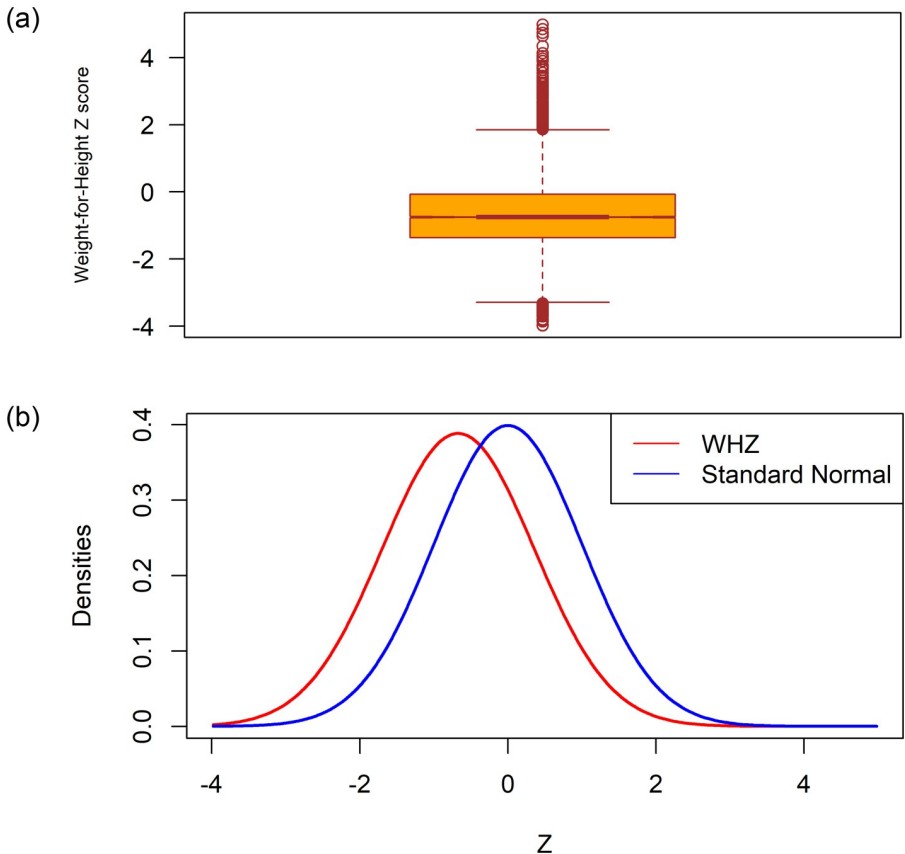

**Fig 1. Box and density plot of the weight-for-height (WHZ) Z-scores with the standard normal variate.**

weighted sample (mathematically adjusted) are available in the published report of BDHS-2017/18 in detail [13]. The weight-for-height Z-score (WHZ) is the target variable, and several child characteristics such as sex, age, duration of breastfeeding, birth order; maternal attributes such as age, maternal educational qualification and BMI; father's education, and attributes related to the child's health are the explanatory variables in this study.

A child is termed wasted if his or her weight-to-height ratio is more than two standard deviations below the reference population's median weight-to-height ratio. This situation indicates an acute nutritional deficiency. This study considers the Z-score of weight-for-height as the target variable. In this paper, the WHZ score is used as the outcome and several characteristics of a child such as sex, age, birth order, breastfeeding length; mother's attributes like age, educational qualification and BMI; factors related to household, and child's health are considered as covariates in this paper. The choice of covariates used in this study was influenced by the availability in the BDHS dataset, self-efficacy as well as guided by existing relevant literature.

## Quantile regression

The quantile regression (QR) model was initially introduced by Koenker and Basset in 1978, and nowadays it is extensively applied in various research areas, particularly in Statistics, Econometrics, and health sciences [34, 36, 42, 53–55]. Suppose, $Y$ be a random (response) variable having cumulative distribution function (CDF) $F_Y(y)$, i.e. $F_Y(y) = P(Y \leq y)$ and $X$ is the p-dimensional vector of predictor variables. Then the $\tau$th (quantile level) conditional quantile

of $Y$ is described as

$$Q_\tau(Y|X = x) = \{y : F_\tau(y|x)\},$$

where $\tau$ varies from 0 to 1. A detailed description of the QR model is presented in S1 Appendix.

We conducted quantile regression for various quantiles (0.1, 0.20, 0.25, 0.5, 0.75 and 0.90) as motivated the literature available in literature. However, in order to determine the importance of performing multiple quantile regressions, we consider the following hypothesis.$H_0$:

$H_0$: all of the estimated coefficients for all quantiles are equal versus

$H_1$: all of the estimated coefficients for all quantiles are not equal.

### Ethics approval

This study was based on an analysis of existing public domain survey data sets that are freely available online with all identifier information removed. The survey was approved by the Ethics Committee of the ICF Macro at Calverton in the USA and by the Ethics Committee in Bangladesh.

## Results

Firstly, we examine the summary statistics of our main target variable i.e., the Z-scores weight-for-height and the findings of WHZ are mean: -0.67, SD: 1.03, skewness: 0.60, and kurtosis: 1.36.

The mean of the WHZ-scores is less than 0 (i.e., a negative mean value for wasting), which indicates that the index's distribution has switched downward and that the majority, if not all, of the children in the population, are malnourished in comparison to the reference group. The coefficient of skewness of the Z-scores depicts that the distribution of the WHZ-scores is slightly positively skewed and fully unmatched with standard normal density and the boxplot presented in Fig 1 revealed that outliers are present in the dataset. Table 1 illustrates the prevalence of wasting among children of under-5 years of age according to selected socio-demographic characteristics. The findings depict that the prevalence of wasting (WHZ-score < -2 SD) in children is about 8 percent and only approximately one percent of children are severely wasted (WHZ-score < -3 SD) in Bangladesh.

Table 1 shows that, in comparison to their male counterparts, female children are slightly less wasted. After the age of two years, the prevalence of wasting reduced, indicating an inverse link between age and the prevalence of wasting. Duration of breastfeeding of child portrayed a positive relationship with the index of wasting. The prevalence of wasting is more frequent among children whose mother's age is less than 18 years compared to others. The prevalence of wasting, a symptom of child malnutrition, is adversely connected to the mother's nutritional status as evaluated by BMI and parental educational qualification, as the prevalence of wasting reduced as the mother's nutrition and parental education levels increased. Results show that non-Muslim children are wasted than their counterparts. Vaccination status and the status of suffering from childhood diseases are also vital in deciding the wasted level of children in Bangladesh. However, in some cases, we observed unexpected results. Moreover, the place of delivery has an impact on a child's nutritional intake like whether they are wasted or not. A child delivered in a health facility has a decreased risk of wasting than a child born at home. A kid's nutritional state is linked to his or her current health status, as a child with diarrhea and fever

**Table 1. Percent distribution of child malnutrition according to wasting by background characteristics.**

| Background characteristics | | Percent | Weight-for-Height (Wasted) in % | | p-value of Chi-square |
|---|---|---|---|---|---|
| | | | Z-score <-3 SD | Z-score <-2 SD | |
| Child's sex | Male | 52.16 | 0.82 | 7.96 | 0.020 |
| | Female | 47.84 | 0.45 | 6.68 | |
| Age of the child | ≤6 months | 13.14 | 0.38 | 1.82 | <0.001 |
| | 7 months-12 months | 9.94 | 0.50 | 4.08 | |
| | 13–24 months | 18.45 | 1.20 | 12.02 | |
| | 25–36 months | 19.87 | 0.97 | 8.06 | |
| | 37–48 months | 19.16 | 0.54 | 7.65 | |
| | 49–59 months | 19.44 | 0.20 | 7.39 | |
| Birth order | 1st | 38.31 | 0.60 | 7.34 | 0.137 |
| | 2nd-3rd | 49.22 | 0.54 | 7.35 | |
| | 4th or higher | 12.46 | 1.13 | 7.16 | |
| Duration of breastfeeding | Never breastfeed | 41.28 | 0.34 | 7.42 | <0.001 |
| | < = 12 months | 2.05 | 0.61 | 7.93 | |
| | 13 or more | 6.97 | 0.95 | 6.27 | |
| | Still breastfeeding | 49.70 | 0.85 | 7.40 | |
| Mother's age | Up to 18 years | 7.23 | 1.73 | 9.67 | 0.008 |
| | 19–24 | 40.24 | 0.44 | 6.86 | |
| | 25–34 | 44.68 | 0.51 | 7.40 | |
| | 35+ | 7.86 | 1.17 | 7.05 | |
| Mother's BMI | Underweight (<18.5) | 13.60 | 1.00 | 12.68 | <0.001 |
| | Normal (18.5–24.9) | 59.21 | 0.67 | 6.94 | |
| | Overweight (> = 25) | 27.18 | 0.34 | 5.41 | |
| Mother's Education level | No Education | 7.15 | 1.06 | 9.73 | 0.056 |
| | Primary | 28.40 | 0.88 | 7.74 | |
| | Secondary and above | 64.45 | 0.47 | 6.89 | |
| Father's Education level | No Education | 14.85 | 0.61 | 8.74 | 0.002 |
| | Primary | 34.29 | 0.75 | 7.31 | |
| | Secondary and above | 50.86 | 0.48 | 6.90 | |
| Type of place of residence | Rural | 73.04 | 0.60 | 7.38 | 0.020 |
| | Urban | 26.96 | 0.72 | 7.24 | |
| Religion | Muslim | 91.96 | 0.65 | 7.43 | 0.291 |
| | Non-Muslim | 8.04 | 0.62 | 6.46 | |
| Place of delivery | With Health Facility | 49.91 | 0.58 | 6.55 | 0.153 |
| | Respondent's Home | 50.09 | 1.02 | 7.88 | |
| Number of ANC visits | None | 13.13 | 0.82 | 7.04 | 0.005 |
| | 1–3 | 44.66 | 0.86 | 7.54 | |
| | 4–7 | 16.18 | 0.76 | 6.91 | |
| | 8 or more | 6.03 | 1.09 | 8.70 | |
| Had diarrhea recently | No | 95.26 | 0.65 | 7.22 | 0.159 |
| | Yes | 4.74 | 0.27 | 9.55 | |
| Had fever in last two weeks | No | 66.79 | 0.54 | 6.24 | <0.001 |
| | Yes | 33.21 | 0.83 | 9.48 | |
| Had cough in last two weeks | No | 64.01 | 0.56 | 6.80 | 0.049 |
| | Yes | 35.99 | 0.76 | 8.28 | |
| Received BCG | No | 6.92 | 0.61 | 2.75 | <0.001 |
| | Yes | 93.08 | 0.83 | 7.58 | |

*(Continued)*

**Table 1.** (Continued)

| Background characteristics | | Percent | Weight-for-Height (Wasted) in % | | p-value of Chi-square |
|---|---|---|---|---|---|
| | | | Z-score <-3 SD | Z-score <-2 SD | |
| Received Vitamin A | No | 30.04 | 0.61 | 5.04 | <0.001 |
| | Yes | 69.96 | 0.88 | 8.19 | |
| Wealth index | Poorest | 21.44 | 0.87 | 8.44 | <0.001 |
| | Poorer | 20.33 | 0.43 | 7.38 | |
| | Middle | 18.86 | 0.93 | 7.15 | |
| | Richer | 19.88 | 0.51 | 7.92 | |
| | Richest | 19.48 | 0.41 | 5.54 | |
| Total | | | 0.64 | 7.98 | |

is more likely to be wasted than a healthy child. Furthermore, the wealth index has significantly positively linked to the wasted level of a child in Bangladesh [Table 1].

The authors test the hypothesis of the equality of coefficients of covariates of Q10 and Q20. It is clear that the p-value of this hypotheses is less than 0.001. As a result, at the 0.1 percent level of significance, the test substantially rejects equality of the estimated coefficients for the quantiles. Now, it can be said that the quintiles with bigger difference will be surely significant. This suggests that in this study, multiple quantile regression approaches are appropriate.

Weight-for-height is thought to be a good short-term indicator of a child's nutritional and health status. Table 2 shows that the WHZ-score at the upper edge of the condition distribution (i.e. after the 50th quantile) is not significant at the 5% significance level, and the score is about half points less in the 90[th] quantile than the 10[th] quantile among female child. Results depict that age of the child is also highly significantly associated with the WHZ score at different quantiles. At different quantiles, the child's age was significantly related to the WHZ score. The WHZ-score increases considerably at all quantiles, i.e. 10[th], 20[th], 25[th], 50[th], 75[th], and 90[th] quantiles, as the mother's BMI improves. Moreover, parental educational status is a significant factor in WHZ score. The results show that current residence has a significant influence on the child's WHZ score at the 10[th], 50[th], and 75[th] quantiles, but the coefficient is significant at the 20[th], 25[th], and 90[th] quantiles. The vaccination and status of suffering from childhood diseases are the contributing factors for determining the level of wasting. Moreover, families economic condition is an indispensable factor in determining the WHZ score of a child [Table 2].

The effect of age of a child is inversely connected to the quantile of WHZ score. The coefficient changes over the range of about 0.07 to approximately 0.046 for the quantile varies from 0.1 to 0.90 for the children whose age lies between 07–12 months compared to children of age less than 06 months. In the case of the birth order of a child, the influence on WHZ score initially increases, however, gradually decreases after the 20[th] quantile. Duration of breastfeeding also shows as an important determinant of wasting of a child and depicts a negative relationship on WHZ score. The age of the mother had a positive and statistically significant relationship with WHZ, however, in some quantile, it shows a negative association; children whose mothers' age lies between 19–24 years age group had a greater WHZ than mothers' who married in their early age (<18 years). The impact of the mother's BMI on the conditional distribution of the WHZ score is statistically significant and fluctuated from quantile to quantile. As for the influence of parental education level, children whose parents had higher educational qualifications are taller than children whose parents are illiterate. Moreover, the findings of quantile regression reveal a positive correlation between household financial status and WHZ score. Children who come from the richest households had 0.163, 0.036, and 0.157 times

**Table 2. Results of quantile regression analysis for WHZ score for under-5 Bangladeshi children.**

| Characteristics | Labels | Q10 Coefficient (95% CI) | Q20 Coefficient (95% CI) | Q25 Coefficient (95% CI) | Q50 Coefficient (95% CI) | Q75 Coefficient (95% CI) | Q90 Coefficient (95% CI) |
|---|---|---|---|---|---|---|---|
| Child's sex | Male (Ref.) | | | | | | |
| | Female | 0.07* (-0.016,0.156) | 0.099** (0.026,0.172) | 0.074* (0.001,0.146) | 0.037* (-0.011,0.085) | 0.022 (-0.052,0.095) | 0.046 (-0.068,0.16) |
| Age of the child | < = 6 months (Ref.) | | | | | | |
| | 7–12 months | -0.062 (-0.376,0.251) | -0.289* (-0.645,0.068) | -0.251* (-0.596,0.095) | -0.272** (-0.486,-0.059) | -0.101* (-0.308,0.107) | 0.234* (-0.155,0.623) |
| | 13–24 months | -0.506*** (-0.844,-0.168) | -0.71*** (-1.006,-0.414) | -0.701*** (-0.992,-0.41) | -0.733*** (-0.917,-0.549) | -0.51*** (-0.741,-0.279) | -0.177* (-0.5,0.146) |
| | 25–36 months | -0.44** (-0.767,-0.113) | -0.699*** (-1.013,-0.385) | -0.706*** (-1.009,-0.404) | -0.807*** (-1.008,-0.606) | -0.708*** (-0.924,-0.492) | -0.492** (-0.878,-0.105) |
| | 37–48 months | 0.35* (0.025,0.676) | 0.14 (-0.246,0.527) | 0.306* (-0.073,0.685) | 0.226* (-0.032,0.484) | 0.231* (-0.047,0.508) | 0.2* (-0.235,0.636) |
| | 49–59 months | -0.35* (-0.833,0.132) | -0.14* (-0.551,0.27) | -0.306** (-0.565,-0.047) | -0.226* (-0.437,-0.015) | -0.231** (-0.448,0.014) | -0.2** (-0.474,0.074) |
| Birth order | 1st (Ref.) | | | | | | |
| | 2nd-3rd | -0.007 (-0.105,0.092) | 0.032* (-0.035,0.099) | -0.02 (-0.083,0.042) | -0.075* (-0.158,0.007) | -0.111** (-0.196,0.026) | -0.152* (-0.334,0.029) |
| | 4th and higher | 0.037* (-0.174,0.248) | 0.046 (-0.099,0.192) | -0.041 (-0.175,0.093) | -0.145** (-0.265,-0.026) | -0.226** (-0.378,0.074) | -0.244** (-0.475,-0.012) |
| Duration of breastfeeding | Never breastfeed (Ref.) | | | | | | |
| | < = 12 months | -0.027 (-0.956,0.903) | 0.031 (-0.877,0.939) | -0.001 (-0.895,0.893) | 0.135 (-0.816,1.085) | -0.178 (-1.509,1.101) | 0.081 (-98.846,99.008) |
| | 13 or more | -0.103 (-1.06,0.854) | 0.03* (-0.87,0.931) | -0.036* (-0.855,0.783) | -0.162* (-1.18,0.857) | -0.414* (-1.673,0.797) | -0.457* (-99.429,98.514) |
| | Still breastfeeding | -0.234* (-1.182,0.715) | -0.084 (-0.997,0.829) | -0.101 (-0.931,0.729) | -0.097 (-1.109,0.915) | -0.281 (-1.494,0.933) | -0.272 (-99.191,98.646) |
| Religion | Muslim (Ref.) | | | | | | |
| | Non-Muslim | 0.011 (-0.121,0.144) | 0.052 (-0.099,0.202) | 0.049 (-0.095,0.193) | -0.01 (-0.146,0.127) | -0.055 (-0.236,0.126) | -0.092 (-0.299,0.115) |
| Mother's age | Up to 18 years (Ref.) | | | | | | |
| | 19–24 | 0.193* (-0.039,0.425) | 0.086** (-0.036,0.208) | 0.067* (-0.054,0.189) | 0.02 (-0.069,0.109) | 0.036 (-0.187,0.259) | 0.008 (-0.242,0.258) |
| | 25–34 | 0.142* (-0.07,0.354) | -0.012 (-0.128,0.104) | -0.018 (-0.146,0.111) | 0.029* (-0.089,0.146) | 0.035* (-0.211,0.28) | -0.026* (-0.323,0.27) |
| | 35+ | 0.09 (-0.273,0.453) | -0.117* (-0.36,0.126) | -0.06* (-0.273,0.153) | -0.031* (-0.205,0.142) | -0.069** (-0.342,0.205) | 0.254** (-0.315,0.823) |
| Mother's Education level | No Education (Ref.) | | | | | | |
| | Primary | -0.054* (-0.259,0.151) | 0.004 (-0.221,0.228) | 0.03 (-0.195,0.256) | 0.051 (-0.13,0.232) | 0.098* (-0.075,0.272) | 0.272*** (-0.044,0.589) |
| | Secondary and above | -0.006 (-0.192,0.18) | 0.04* (-0.156,0.237) | 0.069* (-0.132,0.269) | 0.118** (-0.027,0.264) | 0.064* (-0.106,0.234) | 0.262** (-0.026,0.55) |
| Mother's BMI | Underweight (<18.5) (Ref.) | | | | | | |
| | Normal (18.5–24.9) | 0.206*** (0.052,0.36) | 0.305*** (0.186,0.425) | 0.309*** (0.191,0.426) | 0.292*** (0.194,0.391) | 0.33*** (0.197,0.463) | 0.398*** (0.187,0.609) |
| | Overweight (> = 25) | 0.327*** (0.18,0.473) | 0.417*** (0.304,0.53) | 0.425*** (0.307,0.542) | 0.386*** (0.267,0.504) | 0.596*** (0.439,0.754) | 0.612*** (0.331,0.893) |

(*Continued*)

**Table 2.** (Continued)

| Characteristics | Labels | Q10 | Q20 | Q25 | Q50 | Q75 | Q90 |
|---|---|---|---|---|---|---|---|
| | | Coefficient (95% CI) | Coefficient (95% CI) | Coefficient (95% CI) | Coefficient (95% CI) | Coefficient (95% CI) | Coefficient (95% CI) |
| Father's Education level | No Education (Ref.) | | | | | | |
| | Primary | 0.083 (-0.09,0.257) | -0.005 (-0.148,0.139) | -0.027 (-0.147,0.093) | -0.047 (-0.144,0.049) | -0.002 (-0.12,0.117) | -0.058 (-0.306,0.189) |
| | Secondary and above | 0.087* (-0.048,0.223) | 0.059* (-0.078,0.195) | 0.033 (-0.117,0.184) | -0.003 (-0.118,0.112) | 0.047* (-0.095,0.189) | 0.02 (-0.162,0.203) |
| Type of place of residence | Rural (Ref.) | | | | | | |
| | Urban | -0.077* (-0.172,0.018) | -0.018 (-0.123,0.086) | 0.012 (-0.093,0.116) | 0.049* (-0.035,0.133) | 0.083** (-0.057,0.224) | 0.01 (-0.155,0.175) |
| Place of delivery | With Health Facility (Ref.) | | | | | | |
| | Respondent's Home | -0.003 (-0.099,0.094) | -0.001 (-0.127,0.124) | 0.008 (-0.108,0.125) | -0.004 (-0.086,0.078) | -0.08* (-0.174,0.014) | -0.061* (-0.211,0.089) |
| Number of ANC visits | None (Ref.) | | | | | | |
| | 1–3 | -0.111** (-0.225,0.003) | -0.032* (-0.159,0.094) | -0.047 (-0.175,0.081) | -0.04 (-0.166,0.087) | -0.095* (-0.244,0.055) | -0.21** (-0.398,-0.021) |
| | 4–7 | -0.127* (-0.293,0.04) | -0.016 (-0.14,0.108) | -0.048* (-0.173,0.077) | -0.033* (-0.148,0.083) | -0.099* (-0.252,0.054) | -0.208** (-0.419,0.003) |
| | 8 or more | -0.093 (-0.325,0.14) | 0.024* (-0.212,0.259) | 0.011* (-0.176,0.197) | -0.053* (-0.291,0.185) | -0.029 (-0.267,0.21) | 0.023 (-0.384,0.43) |
| Had diarrhea recently | No (Ref.) | | | | | | |
| | Yes | -0.027 (-0.151,0.097) | -0.06 (-0.219,0.099) | -0.016 (-0.165,0.134) | 0.012 (-0.118,0.142) | -0.055 (-0.246,0.135) | -0.035 (-0.225,0.155) |
| Had fever in last two weeks | No (Ref.) | | | | | | |
| | Yes | -0.226*** (-0.304,-0.147) | -0.207*** (-0.27,-0.144) | -0.209*** (-0.261,-0.156) | -0.212*** (-0.295,-0.129) | -0.21** (-0.349,-0.071) | -0.134** (-0.322,0.053) |
| Had cough in last two weeks | No (Ref.) | | | | | | |
| | Yes | -0.019 (-0.103,0.065) | 0.041 (-0.053,0.135) | 0.04* (-0.032,0.112) | 0.057* (-0.021,0.135) | 0.112* (-0.027,0.251) | 0.073 (-0.157,0.303) |
| Received BCG | No (Ref.) | | | | | | |
| | Yes | 0.027 (-0.169,0.224) | -0.091* (-0.253,0.07) | -0.062 (-0.263,0.138) | -0.138* (-0.333,0.056) | -0.273** (-0.504,-0.041) | -0.653** (-1.235,-0.07) |
| Received Vitamin A | No (Ref.) | | | | | | |
| | Yes | -0.155** (-0.292,-0.018) | -0.094** (-0.172,0.016) | -0.107** (-0.191,-0.024) | -0.168*** (-0.284,-0.053) | -0.29*** (-0.397,-0.183) | -0.353*** (-0.523,-0.184) |
| Wealth index | Poorest (Ref.) | | | | | | |
| | Poorer | 0.132* (-0.008,0.272) | 0.073* (-0.054,0.201) | 0.048 (-0.081,0.176) | -0.067** (-0.181,0.047) | -0.055* (-0.194,0.084) | -0.041* (-0.194,0.113) |
| | Middle | 0.146* (-0.006,0.297) | 0.08* (-0.069,0.23) | 0.1** (-0.055,0.255) | 0.059* (-0.072,0.191) | -0.01 (-0.114,0.094) | 0.024 (-0.162,0.209) |
| | Richer | 0.111* (-0.096,0.319) | 0.038 (-0.08,0.157) | 0.065** (-0.043,0.172) | 0.066** (-0.066,0.197) | 0.029** (-0.112,0.169) | 0.16** (-0.055,0.375) |
| | Richest | 0.163** (0.009,0.317) | 0.036* (-0.097,0.168) | 0.021* (-0.133,0.174) | 0.027* (-0.137,0.191) | 0.09** (-0.164,0.344) | 0.157** (-0.11,0.425) |

Notes: Ref.: Reference category;

*** refers p-value <0.001,

** refers p-value <0.05 and

* refers p-value <0.1.

higher WHZ scores respectively than children who come from the poorest households in the 10th, 20th, and 90th quantiles [Table 2].

The coefficients of selected significant covariates obtained by using a simultaneous quantile regression model ranges 0.10 to 0.90 quantiles of WHZ are presented in Fig 2. The findings reveal that the coefficients vary across quantiles. For example, the coefficient of the female

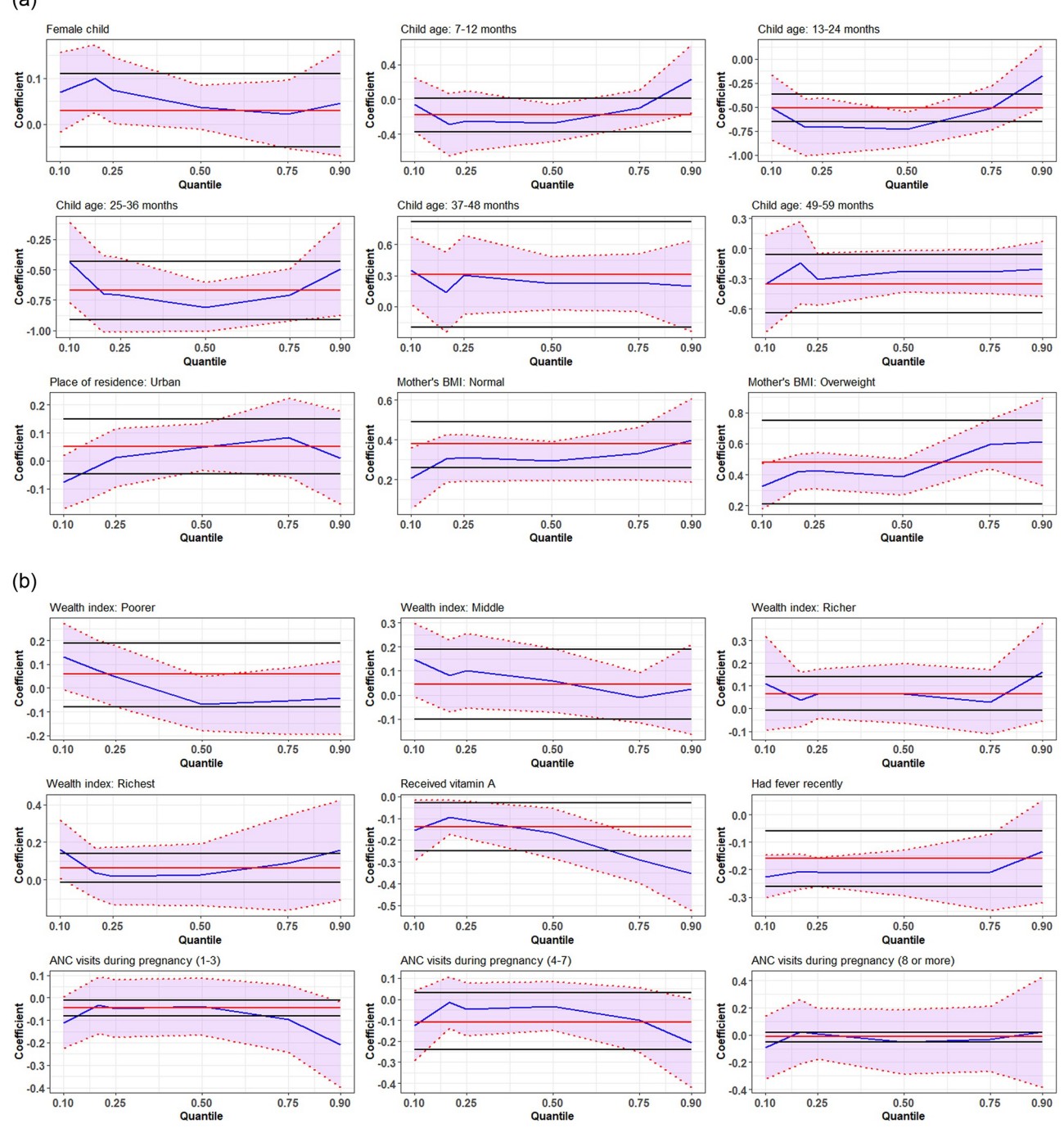

**Fig 2. Plot of covariate effects on quantiles from multivariable simultaneous quantile regression (blue line) and their associated 95% confidence interval (purple shaded regions).** The solid red lines are the ordinary least square regression lines with their 95% confidence intervals (black lines).

child increased from 0.1 quantile to 0.2 quantile, and then it declined up to 0.75 quantile. The coefficients have declined trend up to 0.5 quantile and thereafter an increasing trend is observed for a child aged up to 3 years. The impact of the mother's BMI on WHZ was lower at the lower quantiles of WHZ, however, but higher at the upper tail, i.e., as the quantile increased the coefficient of the mother's BMI is also increased. It is suggesting that children at the lower tail of the WHZ distribution are more likely to be severely wasted than those in the upper end of the distribution. Children from the richest families had a lower risk of under-five severe wasting than the children of the poorest families [Fig 2].

## Discussion

The nutritional status of a child included in the survey is compared to WHO Child Growth Standards, which are focused on a culturally, ethnically, genetically diverse, and internationally representative sample of healthy children living in optimum settings conducive for achieving a child's full genetic growth potential. Wasting is considered as severe if the WHZ score of a child is more than 3 SD (standard deviation) below the reference median. Also, severe wasting is strongly connected to mortality risk. Bangladesh is doing a great job to lessen the burden of wasting over the two decades and a study projected that the prevalence of wasting will decrease by one-quarter by 2030 [56]. There were also significant differences in child nutritional status based on demographic and socio-economic factors. The findings of the quantile regression revealed that the relationship between socio-demographic variables and WHZ varied depending on the conditional WHZ distribution.

The age of a child was found to be strongly connected to wasting in children under the age of five, with the odds of wasting decreasing as the child's age increased, which is consistent with previous research [56, 57]. This could be due to the fact that children are more vulnerable to infections throughout their first year of life [58, 59]. The duration of breastfeeding also shows as an important determinant of wasting of a child and depicts a negative relationship on WHZ score. Breastfeeding is protective against various childhood infectious diseases. The potential reason working behind is that breast milk is the sole natural and primary source of optimum sustenance for newborn babies' physical and neurological growth, and cognitive development, it also boosts the child's immune system at their early age [60–62]. As a result, it is critical to limit the risk of wasting by exclusively breastfeeding up to the first six months of a children's life. The residential status shows a significant relation with WHZ score only at 10th and 20th quartiles in urban areas than rural areas. The environment, choices, employment conditions, social and family networks, access to health care and other services of urbanites differ greatly from those of rural dwellers [63].

Parental education has a significant influence on the level of malnutrition of under-5 children and our findings are consistent with previous studies [40, 42, 57]. The significance of parental education in mitigating the risk of wasting can be explained by enhancing knowledge and understanding about nutrition and health for their children, the possibility of better household income, and the responsibility in food selection decision-making. Moreover, education also offers an opportunity for acquiring protective childcare behaviors such as completing childhood vaccines, improving feeding and sanitation practices, and so on [40, 56, 64]. A study pointed out that higher educational attainment among mothers is effective at promoting women's empowerment and participation in household decision-making, which has been ascertained to lessen malnutrition indices in Bangladesh [65]. To combat malnutrition, programmes for mothers' nutritional education and awareness are essential [11]. Children who suffer from different infectious disease in their life has an impact on wasting. Moreover, the children who consumed vitamin-A and received BCG vaccination had a lower likelihood

of wasting than their counterparts [66]. The findings of the quantile regression exhibited wealth index is statistically significant with the discrepancy in WHZ score. Researchers pointed out that wealthier households can manage to pay for better medical care and additional nutritious food as well as ensure a healthier living environment [40, 56]. The high prevalence of malnutrition is correlated with overall socio-economic deprivation [10].

## Strengths and limitations of this study

The most recent dataset is used in this study and it is a nationally representative cross-sectional survey. This study used quantile regression in order to measure the determinants of malnutrition in Bangladesh. The results from this sample are not transferable to other populations with different characteristics, because socioeconomic factors were only assessed at the one-time point. Moreover, the application of multilevel quantile regression analysis would be a potential topic in a further study that will be helpful to improve the quality of the findings in a future study.

## Conclusion

The effects of covariates included in this study such as a child's age, birth order, parental educational status, and mother's BMI, as well as the status of vaccination and suffering from childhood diseases on child nutrition at various points of the conditional distributions of WHZ-score, were investigated utilizing quantile regressions. Age, mother's BMI, parental educational status, and income index are all major factors of a child's Z score of weight-for-height. To accelerate the reduction of malnutrition or lessen the burden of malnourishment among children by 2030, the authors think that a combined effort should necessarily be taken by the government and public-private owner organizations at the community level. In addition to current programs aimed at improving child health, the government may desire to develop targeted nutrition remediation strategies in order to eliminate childhood malnutrition and prioritize the target group of the population. Furthermore, a healthy mother may give birth to healthy children which indicates the prevalence of undernutrition transfer generation to generation, thus, early intervention programs should not only target children but also their moms in order to improve children's nutritional status. The authors would like to recommend that nutrition and health-related education should be integrated into the educational process in Bangladesh. The authors believe that the findings of this paper will assist policymakers in accelerating the achievement of the SDG-3 in Bangladesh.

## Supporting information

**S1 Appendix.**
(DOCX)

## Acknowledgments

The authors are grateful to ICF International, Rockville, Maryland, USA, for providing the Bangladesh DHS data sets for this analysis. Authors are thankful to the academic editor and two reviewers for their valuable comments and suggestions that help to enhance the manuscript's quality.

## Author Contributions

**Conceptualization:** Md. Moyazzem Hossain, Azizur Rahman.

**Data curation:** Md. Moyazzem Hossain.

**Formal analysis:** Md. Moyazzem Hossain, Faruq Abdulla.

**Methodology:** Azizur Rahman.

**Supervision:** Azizur Rahman.

**Visualization:** Md. Moyazzem Hossain.

**Writing – original draft:** Md. Moyazzem Hossain, Faruq Abdulla.

**Writing – review & editing:** Md. Moyazzem Hossain, Azizur Rahman.

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
