## [Decision Letter · Decision Letter 0]

10 Jun 2022

PONE-D-21-27952Prevalence and Determinants of Wasting of Under-5 Children in Bangladesh: Quantile Regression ApproachPLOS ONE

Dear Dr. Hossain,

Thank you for submitting your manuscript to PLOS ONE. After careful consideration, we feel that it has merit but does not fully meet PLOS ONE’s publication criteria as it currently stands. Therefore, we invite you to submit a revised version of the manuscript that addresses the points raised during the review process. Please see the comments from two reviewers below. The reviewers have provided detailed queries that may be useful to clarify the motivations behind some of the choices made, and should enhance the overall clarity of the manuscript. Please note that there is no requirement to cite any of the specific references suggested by the reviewers.

We look forward to receiving your revised manuscript.

Kind regards,

Hanna Landenmark

Staff Editor

PLOS ONE

Journal Requirements:

Reviewers' comments:

Reviewer's Responses to Questions

**Comments to the Author**

1. Is the manuscript technically sound, and do the data support the conclusions?

Reviewer #1: Partly

Reviewer #2: Yes

2. Has the statistical analysis been performed appropriately and rigorously? 

Reviewer #1: Yes

Reviewer #2: Yes

3. Have the authors made all data underlying the findings in their manuscript fully available?

Reviewer #1: Yes

Reviewer #2: Yes

4. Is the manuscript presented in an intelligible fashion and written in standard English?

Reviewer #1: Yes

Reviewer #2: No

5. Review Comments to the Author

Reviewer #1: Hossain and colleagues examined the prevalence and determinants of wasting among children aged below 5 years in Bangladesh using quantile regression approach based on the 2017/2018 Bangladesh DHS data. This study is critical to understanding the factors associated with the indicator of wasting in the population of this children to inform sound policy decision making. I commend the authors for the application of the quantile regression in their analysis and we look forward to more of this in the literature in relation to modelling nutritional status of under-five children. However, I have some reservations of the use of wasting as the only indicator of nutritional status in this study, the arbitrary use of the tau (i.e., quantiles) values, and why the authors ignored the hierarchical structure of the DHS data in their analysis. See further comments below:

Though the authors attempt to solve an important public health problem, especially in the developing countries like Bangladesh, they failed to justify the use of only wasting as a measure of nutritional status in their study, ignoring other important indicators of nutritional status of children such as stunting (indicator of long-term malnutrition) which is the highest prevalent globally and in developing nations, and underweight among others. Notably, the three commonly used indicators of nutritional status of children below 5 years are stunting, underweight and wasting. Each of this captures different dimension of under-five malnutrition so the authors must provide a scientific reason for choosing only wasting (indicator for short term malnutrition) as the only nutritional status in their study.

Also, like any other DHS data, the Bangladesh DHS data is hierarchical in nature where we have children nested within households, and household nested within clusters (i.e., communities) but the authors did not explain how they account for the hierarchical structure of the data used in this study. Assuming this was not explored during their modelling stage using multilevel quantile regression analysis, it could lead to spurious statistical significance with its associated misleading interpretations. Fortunately, we currently have statistical software packages that allow easy implementation of the multilevel quantile regression analysis. Authors are encouraged to explore this and compare the results for the single level quantile regression to improve the quality of their results in the manuscript.

Furthermore, the arbitrary use of the quantile values is not very informative in this study. Analysing nutritional status indicators using quantile regression should be guided by the thresholds for the quantiles and what they are measuring. For example, a quantile threshold between [0.01, 0.2] measures severe form of stunting, wasting and underweight. Thus, the authors should make conscious efforts to include these thresholds among the selected quantiles analysed and interpret same in relation to the severity of the nutritional status alongside other thresholds outside these to inform sound nutrition policies for these children. They considered 0.1 through 0.9 without any attention to the interpretation in relation to the severity of the wasting based on the quantile regression model. The authors will benefit from the paper by Aheto (2020) below that addressed this issue.

Also, it will be helpful for the readers if the authors provide the plot of the quantile regression coefficients together with the coefficient plot from the ordinary least square regression to allow the comparison between the two approaches as done by Aheto (2020) presented in the reference below.

The discussion and the conclusion look good, but the authors should consider the comments raised above to improve the quality of their manuscript.

Reference

1. Aheto JMK: Simultaneous quantile regression and determinants of under-five severe chronic malnutrition in Ghana. BMC Public Health 2020, 20(1):644. https://bmcpublichealth.biomedcentral.com/articles/10.1186/s12889-020-08782-7

Reviewer #2: Abstract: Strength of associations needs to be reported

Methods:

Lines. Statistical Jargons on quantile regression could be put as a supplementary file.

Lines 162-164 should be placed in method section.

Line 172. Mean, not average.

Line 177. How did you define the outliers? Please mention it clearly somewhere.

Line 179. HAZ is stunting, not wasting. Is it just a typo or the authors coded data incorrectly?

Line 180. Same as above.

Line 185 to 200: Please always use the term that you studied. Malnutrition is a very wide term and the authors explored the factors of wasting only.

Lines 202 to 207. Should be placed in Method section under statistical analysis.

Lines 236-238: Why HAZ again? Please re-write the section carefully and focus on the relationship between the outcome variable and the predictors only.

Discussion:

Lines 17-18: How are the authors so confirm about the confounding effect of infections? Did they test that? If not, it must be properly referred. Biologically, the linear growth spurt slows down with time. As a result wight-for-height becomes more stable with increasing age.

Lines 26-27: It was tough to get the meaning. I don’t know why the authors brought the long-term malnutrition issues here?

Line 35. “Therefore, the government's efforts to…….”- totally redundant.

Line 43: “cross-protective immunity……..an enhanced innate immune response……trained immunity against”- again, totally confusing. The message is not at all clear.

6. PLOS authors have the option to publish the peer review history of their article (what does this mean?). If published, this will include your full peer review and any attached files.

Reviewer #1: **Yes: **Justice Moses Aheto

Reviewer #2: No

---

## [Author Response · Author response to Decision Letter 0]

15 Jun 2022

Dear Editor,

We would like to express our sincere gratitude to the two reviewers and the Academic Editor for their valuable comments. We have considered all the comments made by the reviewers and thoroughly revised and formatted the manuscript accordingly. A detailed response to each of the comments is provided below:

Response to Academic Editor comments:

Thank you very much. The required files are submitted through the submission system. We include all required information in the cover letter.

Response to Journal Requirements:

1. Many thanks. The manuscript is revised according to PLOS ONE’s style. All necessary files are uploaded to the system of the journal

2. Thanks for raising these points. We move the ethical statement in the Methods section. Revised texts are in red color. Page: 6

Response Reviewer 1 comments:

We highly appreciate this comment. Thank you very much. 

Thanks for your in-depth review of the manuscript and potential feedback. We appreciate these comments as they will be helpful to enhance the quality and readability of the manuscript. 

The justification is added in the Introduction section. 

The authors are well known about the three different dimensions used indicators of nutritional status of under-5 children. There are 45.4 million wasted children under the age of five. only more than a quarter of 194 countries are on track to meet the World Health Assembly's (WHA) 2025 target of keeping the prevalence of wasting under 5.0 percent. Moreover, it has the greatest short-term case fatality rate of any form of malnutrition. Revised texts are in red color. Page: 4 

The authors are grateful to the reviewer for highlighting these points. We add this in the limitation section. Revised texts are in red color. Page: 17

Thank you very much for pointing out this issue.

The Results section is revised as per your guidelines. We add the results of 0.2 quantile and we cite the suggested reference. Revised texts are in red color. Page: 7-15

Thanks for your insightful comments. The manuscript is revised accordingly. We add and discuss the plot of the quantile regression coefficients. Revised texts are in red color. Page: 7-15

Thanks for your positive comments. It motivates us. 

Response to Reviewer 2 comments:

Thank you very much for your valuable comment and suggestions that help us improve the manuscript's quality. We have revised the Abstract section. Revised texts are in red color. Page: 1

Thanks. The title is revised as per your comment. 

We move the Statistical Jargons on quantile regression in Appendix 1. 

We move Lines 162-164 in the Methods section. 

Line 172 is revised.

Line 177 is revised as per your comment.

Line 179-180, it was a typo. You are right. We revise it. 

Thanks. Line 185-200 is revised as per your comment.

Lines 202-207 are placed in the Methods section. 

Thanks. We revise typos in Lines 236-238. Revised texts are in red color. Page: 5-10 

We appreciate the feedback. We do not test the effect of infections. We add references in Lines 17-18. 

Lines 26-27 are revised. 

Line 35 is deleted because of redundancy. 

Line 43 is deleted. Revised texts are in red color. Page: 15-16

Finally, the revised manuscript has been produced following the valuable comments and suggestions of the reviewers. Once again, we would like to thank the reviewers for their sincere dedication, professional insights, and earnest cooperation in reviewing the manuscript.

---

## [Decision Letter · Decision Letter 1]

25 Oct 2022

PONE-D-21-27952R1Prevalence and Determinants of Wasting of Under-5 Children in Bangladesh: Quantile Regression ApproachPLOS ONE

Dear Dr. Hossain,

Thank you for submitting your manuscript to PLOS ONE. After careful consideration, we feel that it has merit but does not fully meet PLOS ONE’s publication criteria as it currently stands. Therefore, we invite you to submit a revised version of the manuscript that addresses the points raised during the review process.

We look forward to receiving your revised manuscript.

Kind regards,

Saqlain Raza

Academic Editor

PLOS ONE

Journal Requirements:

Additional Editor Comments (if provided):

1. In the abstract, details about the data collection is not mandatory. In the manuscript, it is unnecessarily lengthy. Authors can reduce methods and materials in the abstract.

2. The 'Conclusion' in the abstract needs to be more specific with the results. Authors need to point which government or pubic-private organizations can make the situation better.

3. In Table 2, authors have tested the quantiles. It is understood that if the two closer quantiles are statistically significant, their successors will be surely significant, too. In the table, two quintiles Q10 and Q20 are significant. It means the quintiles with bigger difference will be surely significant. What is the reason to include so many quintiles in the table? If authors believe that these are redundant, they may remove the additional quintile results from the table. The results are sufficiently communicated in the interpretation.

4. The authors are suggested to replace the old references with the new one. For example, a wide literature is available in recent years:

https://journals.plos.org/plosone/article?id=10.1371/journal.pone.0263470

https://www.frontiersin.org/articles/10.3389/fpubh.2022.792164/full

It provides the healthy discussion comparing different methods of measurements.

Reviewers' comments:

Reviewer's Responses to Questions

**Comments to the Author**

1. If the authors have adequately addressed your comments raised in a previous round of review and you feel that this manuscript is now acceptable for publication, you may indicate that here to bypass the “Comments to the Author” section, enter your conflict of interest statement in the “Confidential to Editor” section, and submit your "Accept" recommendation.

Reviewer #1: (No Response)

Reviewer #2: All comments have been addressed

2. Is the manuscript technically sound, and do the data support the conclusions?

Reviewer #1: Yes

Reviewer #2: Yes

3. Has the statistical analysis been performed appropriately and rigorously? 

Reviewer #1: Yes

Reviewer #2: Yes

4. Have the authors made all data underlying the findings in their manuscript fully available?

Reviewer #1: Yes

Reviewer #2: Yes

5. Is the manuscript presented in an intelligible fashion and written in standard English?

Reviewer #1: Yes

Reviewer #2: Yes

6. Review Comments to the Author

Reviewer #1: I thank the authors for the revision, good job.

The revision made by the authors are satisfactory. However, there are few things to correct. For example, in Table 3 column 4, the authors stated Q25 instead of Q20, leading to duplication of Q25 in the table.

In addition, the Figure 2 provided as response to my earlier query improved the understanding of the effect of covariates on the various quantiles of weight for height z-score. However, the authors forgot to add the ordinary least squares regression coefficient line to the plot which I recommended in my earlier review to put the quantile regression in perspective. This will show why the quantile regression approach is preferred to the ordinary least squares regression approach. I provided an example based on a published paper in my first review (see below here again) to guide the authors. In that paper, the authors will find a solid red lines in Figure 2 which represent the ordinary least squares regression line. Same was stated beneath the Figure 2 in the published paper.

Reference

Aheto JMK: Simultaneous quantile regression and determinants of under-five severe chronic malnutrition in Ghana. BMC Public Health 2020, 20(1):644. https://bmcpublichealth.biomedcentral.com/articles/10.1186/s12889-020-08782-7

The authors made me spent much time to review their revision because they did not do their rebuttal letter well. They are expected to respond to each item point-by-point, but in their case, they only provided responses without referring to the queries. They should remember to do this in their next revision.

Please, once the above are addressed, the paper should be sound for publication.

Reviewer #2: (No Response)

7. PLOS authors have the option to publish the peer review history of their article (what does this mean?). If published, this will include your full peer review and any attached files.

Reviewer #1: **Yes: **Justice Moses Aheto

Reviewer #2: No

---

## [Author Response · Author response to Decision Letter 1]

4 Nov 2022

Dear Saqlain Raza

Academic Editor

PLOS ONE

We would like to express our sincere gratitude to the two reviewers and the Academic Editor for their valuable comments. We have considered all the comments made by the reviewers and thoroughly revised and formatted the manuscript accordingly. A detailed response to each of the comments is provided below.

Response to the Academic Editor comments:

Thank you very much. The required files are submitted through the submission system. 

Response to the Journal Requirements:

Many thanks. We check all the references and ensure that all are correct and complete. 

All necessary files are uploaded to the system of the journal. 

Response to the Additional Editor Comments:

1. Thank you very much for your comment and feedback. We revised the abstract as per your comment. Revised texts are in red color. 

Page: 1

2. Thanks. The conclusion in the abstract is revised. Revised texts are in red color. 

Page: 2

3. Thanks. We revised the manuscript and Remove Table 2. Revised texts are in red color. 

Page: 9

4. Thanks. We revised the Introduction and Discussion sections and cite the suggested papers (Ref. 10, Ref. 11). Revised texts are in red color. 

Page: 3, 16

Response to the Reviewer 1 comments:

1. We highly appreciate this comment. It was a typo. We revise it. 

Revised texts are in red color. Page: 11-13

2. Thanks for your insightful comments. Figure 2 is revised accordingly. 

We also cited the suggested paper (Ref. 42). Revised texts are in red color. 

Page: 14 

3. Thanks. We add the point-by-point author’s response file along with the revised version. 

We are thankful to the reviewer for providing comments and feedback. The authors are also grateful to the reviewer for recommending publication after the revision of the manuscript. We believe that it helps to improve the quality of the manuscript. 

Finally, the revised manuscript has been produced following the valuable comments and suggestions of the reviewers. Once again, we would like to thank the reviewers for their sincere dedication, professional insights, and earnest cooperation in reviewing the manuscript.

---

## [Editor Report · Decision Letter 2]

10 Nov 2022

Prevalence and Determinants of Wasting of Under-5 Children in Bangladesh: Quantile Regression Approach

PONE-D-21-27952R2

Dear Dr. Hossain,

We’re pleased to inform you that your manuscript has been judged scientifically suitable for publication and will be formally accepted for publication once it meets all outstanding technical requirements.

Kind regards,

Saqlain Raza

Academic Editor

PLOS ONE

Additional Editor Comments (optional):

Remark: The authors added some suggested studies in their manuscript. But it seems that they only relied on the results of these studies and mentioned in their manuscript. I would urge the authors to read the methodology for more clarity on the topic and for the future research.
---

## [Editor Report · Acceptance letter]

14 Nov 2022

PONE-D-21-27952R2 

Prevalence and Determinants of Wasting of Under-5 Children in Bangladesh: Quantile Regression Approach 

Dear Dr. Hossain:

I'm pleased to inform you that your manuscript has been deemed suitable for publication in PLOS ONE. Congratulations! Your manuscript is now with our production department. 

Kind regards, 

on behalf of

Dr. Saqlain Raza 

Academic Editor

PLOS ONE